# Effect of Bi on the Performance of Al-Ga-In Sacrificial Anodes

**DOI:** 10.3390/ma17040811

**Published:** 2024-02-08

**Authors:** Xin Liu, Yufeng Lin, Yu Li, Nian Liu

**Affiliations:** 1Department of Basic, Naval University of Engineering, Wuhan 430033, China; liuxin2008dragon@126.com (X.L.); 18163558261@163.com (Y.L.); 2Navy 91844 Troops, Guangzhou 510000, China

**Keywords:** sacrificial anode, aluminum alloy, electrochemical properties, microstructure SVET

## Abstract

Cathodic protection is widely used for metal corrosion protection. To improve their performance, it is necessary and urgent to study the influence of metal oxides on the microstructure and performance of aluminum alloy sacrificial anodes. Taking an Al-Ga-In sacrificial anode as the research object, the dissolution morphology and current efficiency characteristics were studied by means of electrochemical testing and microstructural observation, and the influence of varying Pb and Bi contents on the performance of an aluminum alloy sacrificial anode was investigated. The test results reveal that: (1) The Al-Ga-In sacrificial anode with 4% Pb and 1% Bi contents exhibits the best sacrificial anode performance. (2) The inclusion of an appropriate Bi element content shifts the open-circuit potential in a negative direction and promotes activation dissolution. Conversely, excessive Bi content leads to uneven dissolution, resulting in the shedding of anode grains and greatly reducing the current efficiency. (3) During the activation dissolution of the aluminum alloy, the second phase preferentially dissolves, and the activation point destroys the oxide film, resulting in the dissolution of the exposed aluminum matrix. Consequently, the concentration of dissolved metal ions is reduced and deposited back on the surface of the anode sample, promoting the continuous dissolution of the anode.

## 1. Introduction

In marine engineering, the corrosion of metal materials by seawater greatly reduces the utilization efficiency, service life, and safety coefficient of the materials [1,2,3]. Various anti-corrosion measures have been set in place, among which the sacrificial anode cathodic protection method is one of the most economical and effective anti-corrosion methods [4,5]. Sacrificial anode materials mainly include magnesium, zinc, and aluminum alloys. Aluminum alloys have emerged as the preferred sacrificial anode materials, gradually superseding traditional zinc anodes, owing to their abundant resources, cost-effectiveness, substantial theoretical capacitance, and rapid advancements in marine applications.

Research has demonstrated that adding Pb, Bi, In, Ga, and other alloy elements into aluminum alloys causes a negative potential shift, improving the efficiency of the sacrificial anodes [6,7,8]. Particularly, the Pb and Bi alloy elements significantly impact the aluminum anodes. For example, Li Xiaoxiang et al. have shown that Pb exhibits a notable hydrogen desorption overvoltage [9]. The enrichment and physical activation on the aluminum surface, coupled with the melting and shedding of low-melting-point compounds within the alloy, contribute to a reduction in electrochemical and resistance polarization. This process enhances the electrochemical activity of the material; hence, the aluminum alloy anode material achieves a highly negative stable potential and exhibits a low self-corrosion rate. Guo Jianzhang et al. have demonstrated that adding an appropriate amount of Bi effectively destroys the oxide film on the aluminum alloy anode surface, improves the anode activation performance, reduces the effect of grain boundary corrosion, and improves the dissolution morphology of the anode [10]. Consequently, the strengthening effect of Pb and Bi on the aluminum alloy sacrificial anodes and the influence of the microstructure on aluminum alloy anode performance have become research hotspots. Several studies have shown that Pb and Bi have synergistic effects, improving the active dissolution and current efficiency of aluminum alloys. Moreover, studies have demonstrated that the total content of both elements should not exceed 5% [11]. However, no in-depth study on Al-Ga-In alloy sacrificial anodes has been conducted yet. Therefore, the present study takes Al-Ga-In alloy sacrificial anodes as the research object. A combination of electrochemical tests and microstructure observations is employed to analyze the performance of these anodes. Moreover, this study investigated the dissolution morphology and current efficiency characteristics by determining the impact of Pb and Bi content on the performance of aluminum alloy sacrificial anodes. The findings aim to offer valuable data support for practical applications.

## 2. Experimental Methods

As per the literature [12], optimal active dissolution and current efficiency are observed when the total content of both elements, namely Pb and Bi, is approximately 5%. Ga and In can improve alloy dissolution activity and enhance current efficiency, and their contents should not be too high. Their reasonable content is 0.01% and 0.025%, respectively. Therefore, in this experiment, a certain amount of Ga and In were added to pure aluminum to obtain an Al-Ga-In aluminum alloy. On the basis of the Al-Ga-In aluminum alloy, the total content of Pb and Bi was set to 5%. The influence of the synergistic effect of the two elements on the sacrificial anode material was investigated by adjusting the elemental contents of Pb and Bi. Table 1 illustrates the sample labels and composition formulas.

The aluminum alloys discussed in the paper were all prepared in the laboratory. The melting equipment was a vacuum induction suspension melting furnace, produced by Shenzhen Saimaite New Materials Co., Ltd. in Shenzhen, China, with the model number XF-1. Firstly, we placed the pure aluminum ingot in a graphite crucible and heated it in a vacuum induction suspension melting furnace. After the aluminum ingot was completely melted, we added our pre-weighed alloy elements (such as Pb, Bi, Ga, and In). Subsequently, we controlled the induction current to suspend the metal melt so that all elements were fully mixed. We poured the metal melt into the mold at an appropriate speed to form a round rod of a certain size and performed subsequent processing according to different test requirements. The use of vacuum induction suspension melting eliminates the influence of environmental media and ensures uniform mixing, resulting in reliable aluminum alloy composition.

### 2.1. Electrochemical Performance Test

The samples and experimental devices were prepared according to the GB/T17848-1999 [13] test methods for the electrochemical performance of sacrificial anodes. The auxiliary cathode consisted of cylindrical carbon steel; the reference electrode was a saturated calomel electrode; the working electrode contained an internal and external surface; and the working electrode area was $840\;{\mathrm{cm}}^{2}$. The welded area between the auxiliary cathode and the copper rod was sealed with wax. The open-circuit potential of the anode was measured within 1 h after immersing a sample of the aluminum anode in a sodium chloride solution. A sliding rheostat was adjusted to maintain a constant current of $1\;\mathrm{mA}/{\mathrm{cm}}^{2}$. The working potential of the anode was measured every 24 h for 10 days. Then, the circuit was disconnected, the samples and the copper coulometer were removed, the corrosion pattern was recorded, and the weight changes of the anode samples and cathode copper sheets were calculated. Moreover, the anode current efficiency was calculated.

### 2.2. Electrochemical Behavior Test Methods

A CORRTEST CS310 electrochemical workstation was used to assess the corrosion resistance of the sample, and supporting test software was used to measure the linear polarization curve and electrochemical impedance spectrum. Epoxy resin was used to seal the edges of the sample, leaving a working surface with an area of 1 cm^2^, and the sample was fixed in a test container. A three-electrode system was used in the experiment; the working electrode corresponded to the sample to be tested, the auxiliary electrode was a Shanghai Thundermagnetic 213 platinum electrode, and the reference electrode was a Shanghai Thundermagnetic 232 saturated calomel electrode. A 3.5% NaCl solution was used as the experimental electrolyte. The experimental device was kept in a terrestrial atmospheric environment, and the experimental temperature was set to 25 °C. Before starting the experiment, the sample was polished to 1000 grit, cleaned with absolute alcohol, and assembled. Once the open-circuit potential stabilized, the Tafel polarization curve and electrochemical impedance spectrum were measured. The scanning range of the Tafel polarization curve was −0.1~0.1 V (relative to the open-circuit potential); the scanning rate was 0.5 mV/s; and the frequency range of the electrochemical impedance spectrum (EIS) was 0.01~100 KHz.

### 2.3. Microstructure Observation and Composition Analysis

The prepared sample was cleaned with anhydrous ethanol and manually ground with sandpaper until no obvious abrasion marks could be observed. Then, electrolytic polishing corrosion was performed with a 30% KOH solution. The electrolytic voltage was set to 0.2 V, whereas the polishing time was set to 2 min. After that, the prepared samples were cleaned and dried. Finally, the microstructure was examined using a VDX 5000 3D which was made by KEYENCE in Tokyo Japan and a Leica DM 2500M metallographic microscope which was made by LEICA in Heerbrugg Switzerland. Before and after the electrochemical experiment, the sample surface was cleaned and dried with anhydrous ethanol and then put into an AURIGA scanning electron microscope to observe its microscopic morphology. Moreover, an energy-dispersive spectrometer (EDS) equipped with a scanning electron microscope (SEM) was used for element energy spectrum analysis.

### 2.4. Scanning Vibrating Electrode Test (SVET)

A Princeton microarea electrochemical testing system, VersaSCAN, was used to measure the potential gradient caused by the local current on the surface of the aluminum alloy sacrificial anode, as well as to describe the surface activation dissolution process of the aluminum alloy anode samples and to explore the corrosion dissolution mechanism of the Pb and Bi elements on the sacrificial anode.

## 3. Results and Discussion

### 3.1. Microstructural Observation and Compositional Analysis

The microstructure of the specimen was examined using an optical microscope, and its structural arrangement is shown in Figure 1. The microstructure of the alloy mainly consists of a gray matrix and a black granular or reticular structure. The gray matrix represents the substrate, the black particles are the second phase of segregation, and the black reticular structure is the grain boundary after erosion.

When only the Pb element is present, sample A1 exhibits a higher concentration of black particles as well as a black network structure, displaying an uneven distribution of microstructure and significant intergranular corrosion. When the amount of Bi is added, the microstructural distribution of samples A2, A3, and A4 is relatively uniform, whereas the degree of intergranular corrosion is significantly weakened. However, when the Bi content is too high, sample A6 suffers from severe grain boundary segregation and significant intergranular corrosion. This illustrates that adding an amount of Bi ranging from 1% to 3% can effectively reduce the effect of grain boundary corrosion and promote the uniformity of anodic dissolution. In samples A2, A3, and A4, the inclusion of Bi facilitates a more uniform distribution of the structure. This can mitigate the pronounced self-correction resulting from the initial segregation phase, thereby ensuring current efficiency.

### 3.2. Electron Probe Scanning Analysis

For samples A2, A3, and A4, electron probe technology was used to further analyze the distribution of Bi elements with different contents. Figure 2, Figure 3 and Figure 4 represent the electron probe scanning pictures of the distribution of the Al and Bi elements in samples A2, A3, and A4.

A more uniform distribution of Bi in the alloy contributes to better structural uniformity and enhanced dissolution performance and stability during the anodic dissolution process. The electron probe scanning image reveals bright white spots, primarily corresponding to the aggregation areas of the Bi element. These spots serve as activation points in the anodic dissolution process. The bright white spots in sample A2 have a wider area and are more evenly distributed. Therefore, A2 exhibits a relatively higher number of activation points, facilitating easier activation and dissolution. It is speculated that its dissolution is more stable, leading to higher current efficiency.

### 3.3. Work Performance Analysis

Operational performance primarily encompasses the working potential of the sacrificial anode, the actual electric capacity, and the current efficiency. The working performance parameters are shown in Table 2.

The results in Table 2 demonstrate that A3, A4, and A5 have more negative potentials in the open-circuit potential, consistent with previous microstructure analysis results. Due to the presence of more active spots in the microstructure, intergranular corrosion is likely to occur, resulting in a more negative potential. However, although A3, A4, and A5 exhibit relatively negative working positions, the working potential displays significant fluctuations. This indicates an unstable electrode surface and uneven anode dissolution, which may lead to material detachment and reduced current efficiency. The data on current efficiency further suggests that all three samples exhibit a very low current efficiency, falling below 40%. The corrosion morphology of the samples after a 30-day electrochemical performance test is illustrated in Figure 5, and the dissolution morphology is detailed in Table 3. The results demonstrate that the surface corrosion of samples A3, A4, and A5 is uneven, with material detachment occurring during the experiment. This phenomenon indicates a reduction in the current efficiency of the sacrificial anode. Moreover, it suggests that the Pb and Bi content ratio is inappropriate at this stage, leading to excessively rapid dissolution.

The sacrificial samples A1 and A6, with only Pb or Bi added based on their heterogeneity, had relatively positive working potentials, below −1.0 V. This is likely because Pb, being a high hydrogen overpotential element, facilitated a higher overpotential of the cathodic phase hydrogen precipitation reaction. Simultaneously, the hydrogen depolarization reaction of the cathodic phase impurities was suppressed, hindering the dissolution process of the aluminum anode. This reduction in the self-corrosion rate improves the current efficiency of the aluminum anode. When Pb and Bi coexist, the open-circuit potential of the aluminum anode is more negative. Pb and Bi exhibit synergistic effects, and the addition of an appropriate content of Pb and Bi promotes a negative shift in the open-circuit potential of the aluminum anode alloy. The A2 sample, containing an appropriate Pb and Bi content, has a relatively negative working potential of approximately −1.1 V and a higher current efficiency of nearly 80%. Therefore, A2 facilitates an excellent performance of the sacrificial anode, exhibiting a moderate open-circuit potential, uniform activation and dissolution, and a relatively high current efficiency.

### 3.4. Polarization Curve Analysis

Polarization curves are commonly employed to elucidate the fundamental principles of metal corrosion, reveal the underlying mechanisms of metal corrosion, and investigate strategies for corrosion control. Figure 6 represents the anodic polarization curve of the aluminum alloy specimen, whereas Table 4 contains the Tafel fitting data for the anodic polarization curve.

Figure 6 shows that the anodic polarization curves of samples A1, A2, and A6 are similar and relatively smooth. Moreover, there is no obvious passivation phenomenon, indicating that the three samples have good activation performance in a 3% NaCl solution. Importantly, the three polarization curves present two obvious discharge peaks, corresponding to two discharge processes. According to the Tafel fitting results in Table 4, the self-corrosion current density, as well as the corrosion rate of the three anodes, are relatively small. However, the anodic polarization curves of samples A3, A4, and A5 are similar, exhibiting current plateau areas. In these plateau areas, variations in the anode potential have very little effect on the current change, and small current changes may result in substantial potential fluctuations. The formation of current plateaus may be attributed to the relatively high resistance of the passive film on the anode surface, making it challenging to break down when the potential is negative. When the potential exceeds a certain value, the passive film can break down, and the anode current increases rapidly. The Tafel fitting results presented in Table 4 indicate that the self-corrosion potential of samples A3, A4, and A5 is negative, while the self-corrosion current density is significantly larger, being two orders of magnitude higher than that of A1, A2, and A6. A significant number of hydrogen evolution reactions occurred during the test, generating a large number of bubbles on the anode surface. This led to grain shedding in the anode, ultimately reducing the current efficiency. Therefore, considering the corrosion rate, A3, A4, and A5 are unsuitable for sacrificial anode materials.

Among samples A1, A2, and A6, sample A2 exhibited the lowest corrosion rate and the most negative self-corrosion potential. Conversely, while sample A1 actively dissolves and serves as a sacrificial anode to protect the cathode, its lower corrosion rate indicates more even corrosion, reducing anode material shedding and improving current efficiency. This aligns with the observed trend in the change in current efficiency in the working performance of sacrificial anodes analyzed previously.

Based on the compositional analysis, the samples range from A1 to A5, reflecting a gradual decrease in the Pb element and a gradual increase in the Bi element. As the Bi element gradually increased, the potential shifted negatively, and the corrosion current density initially decreased and then increased. In general, the dissolution morphology worsened, leading to a reduction in current efficiency. Therefore, when Pb and Bi coexist, the optimal addition amount of Bi is 1.0%, indicating that A2 exhibits the best sacrificial anode performance.

### 3.5. EIS Analysis

The EIS of the aluminum alloy anode was used to perturb the system with small-amplitude electrical signals to obtain more kinetic and electrode interface structure information. Figure 7 shows the EIS of the aluminum alloy anode. Due to the smaller EIS radii of A3, A4, and A5, they cannot be fully displayed in Figure 7a, which is the area shown by the green circle in the figure. Enlarge the display of this area in Figure 7b. Therefore, Figure 7a shows the electrochemical impedance spectra of A1, A2, and A6, while Figure 7b shows the electrochemical impedance spectra of A3, A4, and A5.

The parameters of the electrochemical impedance spectra were fitted according to an equivalent circuit diagram (Figure 8) and are presented in Table 5.

The EIS of each sample comprises two capacitive arcs. The fitting data indicates that, with a decrease in Pb content and an increase in Bi content, the electric double-layer capacitance between the metal/passive film and the solution initially increases and then decreases. Simultaneously, the polarization resistance first decreases and then increases.

By analyzing different alloy compositions, it can be found that: (1) A6 does not contain Pb, only 5% Bi. A5 contains 1% Pb and 4% Bi, while A2, A3, A4, and A5 contain a certain amount of both Pb and Bi. (2) Compared to the samples containing Bi or Pb alone (A1 and A6), the simultaneous addition of Bi and Pb (samples A2, A3, A4, and A5) may increase or decrease the corrosion rate of the alloy, depending on the synergistic effect of Bi and Pb. (3) By comparing the corrosion rates of A2, A3, A4, and A5, it was found that when the Bi and Pb contents were 4% and 1%, respectively (sample A2), the corrosion rate was the lowest and the impedance spectrum radius was the largest.

Alloy A2 (Al-4Pb-1Bi) exhibits the largest electric double-layer capacitance and the smallest polarization resistance, representing the lowest impedance value. This suggests that A2 is the most susceptible to corrosion and dissolution, aligning with the findings of the working performance analysis and polarization curve test. The results suggest that the polarization resistance of A2 is moderate, facilitating a corrosion dissolution reaction that is not excessively corrosive. This ensures that serious self-corrosion does not occur during the continuous activation and dissolution processes, ultimately improving current efficiency. Consequently, the performance of sample A2 is superior, representing the compositional design with the best overall electrochemical performance for aluminum alloys.

### 3.6. SVET Analysis

The above analysis suggests that alloy A2 (Al-4Pb-1Bi) has excellent sacrificial anode performance. To further investigate this observation, a scanning vibrating electrode was employed to examine the pitting corrosion of the double electric layer of alloy A2. The focus was placed on examining the micro-area electrochemical performance during the destruction and repair of the passive film with the aim of analyzing the corrosion behavior impacting the entire material and exploring the corrosion process and mechanism of pitting corrosion and passive film destruction. Figure 9, Figure 10 and Figure 11 show three-dimensional and two-dimensional plane views of the SVET test of sample A2 after soaking it in a sodium chloride solution (3.5% mass fraction) for 1.5 h, 3.5 h, and 5.5 h, respectively.

Figure 9 shows that after the sample was soaked for 1.5 h, an area with a significant potential difference appeared in the middle of the sample. The size of this area corresponds to approximately 2.5 mm × 2.5 mm. The second-phase particles dissolve preferentially, leading to the rupture of the passive film and thus changing the ion current above the region. After the passive film is partially broken, the aluminum substrate is exposed and starts to activate and dissolve, acting as an anode.

Figure 10 shows that the original peaks in the fourth and fifth columns of the first row and the second and third columns of the second row of the two-dimensional image disappear with time, while new peaks appear in the first column of the second row of the grid. It can be assumed that the potential of the aluminum anode gradually becomes negative as the aluminum substrate continues to be activated and dissolved. When the potential drops to a certain point, the previously dissolved alloying element ions are reduced and deposited back onto the anode surface, forming new cathodic phases, as confirmed by the appearance of new positive peaks, as illustrated in Figure 10. These newly formed cathodic phases can serve as activation points, consistently facilitating anodic activation and dissolution. The peaks in the first row and column persist, albeit with slight upward shifts and expanded mapping regions. It is hypothesized that the rupture of the oxide film in this region is followed by the chemical dissolution of the surrounding aluminum matrix. The second-phase particles in this area are slightly larger, gradually revealing the shape of the second phase.

With increasing corrosion time, the continuous aggregation of anions and cations leads to a gradual increase in the potential difference between the second interaction and the matrix, indicating that the corrosion activation process is deepening. Figure 11 shows that after immersion for 5.5 h, the maximum potential difference in this area is more than nine times that of the first scan and more than two times that of the second scan. The blocky, warm color areas in the fourth and fifth columns of the second row indicate that a large number of metal ions are deposited back to form a cathode phase. A warm-colored area is always present in the first row and column. The area gradually expands during the scanning process, and the complete shape of the warm-colored area is not visible in the scanning area. This is because the second-phase particles in this region are larger, have more contact with the substrate, and are challenging to dissolve or separate.

In summary, the activation and dissolution processes of the aluminum alloy can roughly be described as follows: (1) starting from the preferential dissolution of the second phase, as the reaction proceeds, the passivation film undergoes a local fracture, the aluminum substrate is exposed, and activation and dissolution begin; (2) as the reaction continues, the small-sized second phase is dislodged and dissolved, and the potential of the aluminum anode is shifted. After the metal ions dissolved in the medium are deposited back to the anode surface, a new cathodic phase is formed, serving as a new activation point for the continuous dissolution of the aluminum anode. (3) If there is a more negative second phase or inclusions in the alloy, severe pitting corrosion will occur.

## 4. Conclusions

The Al-Ga-In sacrificial anode has the best sacrificial anode performance when 4% of Pb and 1% of Bi are added.

An appropriate Bi element content can shift the open-circuit potential negatively and promote activation dissolution. Excessive Bi elements are prone to uneven dissolution, resulting in the shedding of anode grains and greatly reducing the current efficiency.

During the activated dissolution of aluminum alloys, when the second phase preferentially dissolves or the activation point destroys the oxide film, the initial dissolution exposes the aluminum substrate. Subsequently, the already dissolved metal ions are reduced and deposited back onto the surface of the anode sample, facilitating the continuous dissolution of the anode.

## Figures and Tables

**Figure 1 materials-17-00811-f001:**
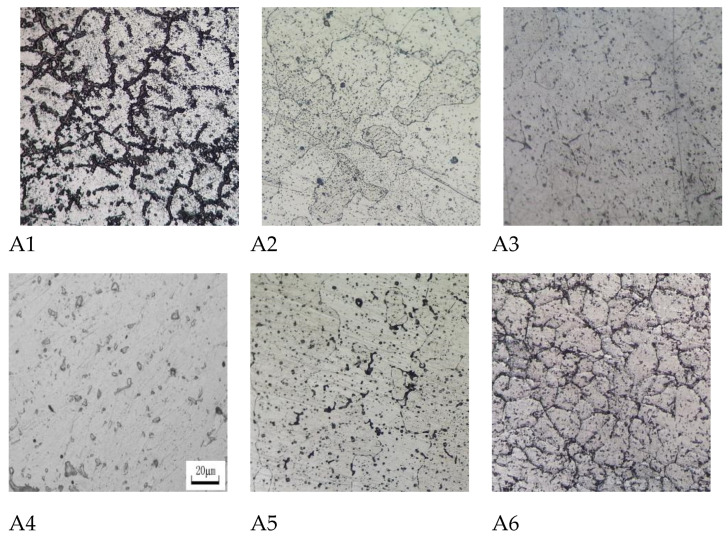
Metallographic structural arrangements of the specimen.

**Figure 2 materials-17-00811-f002:**
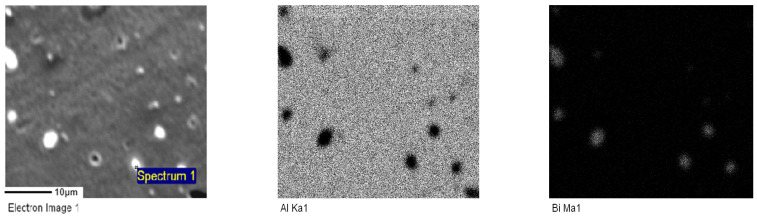
Element surface scanning diagram of aluminum anode sample A2.

**Figure 3 materials-17-00811-f003:**
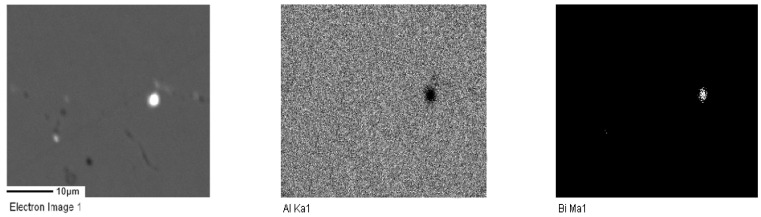
Element surface scanning diagram of aluminum anode sample A3.

**Figure 4 materials-17-00811-f004:**

Element surface scanning diagram of aluminum anode sample A4.

**Figure 5 materials-17-00811-f005:**
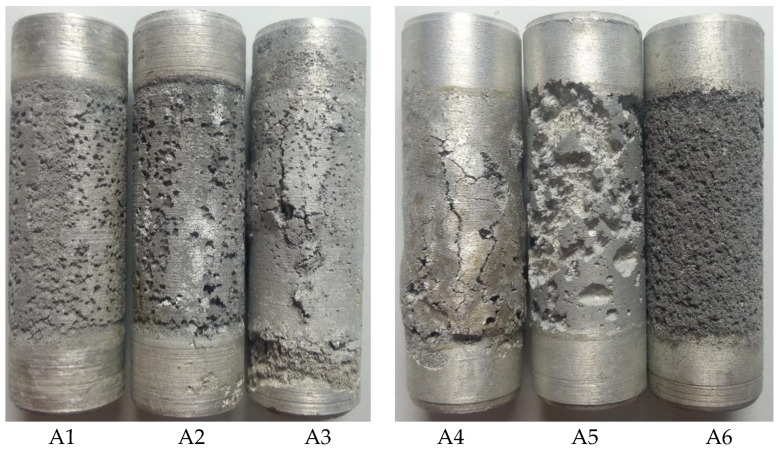
Anode surface morphology after a 240-h constant current test.

**Figure 6 materials-17-00811-f006:**
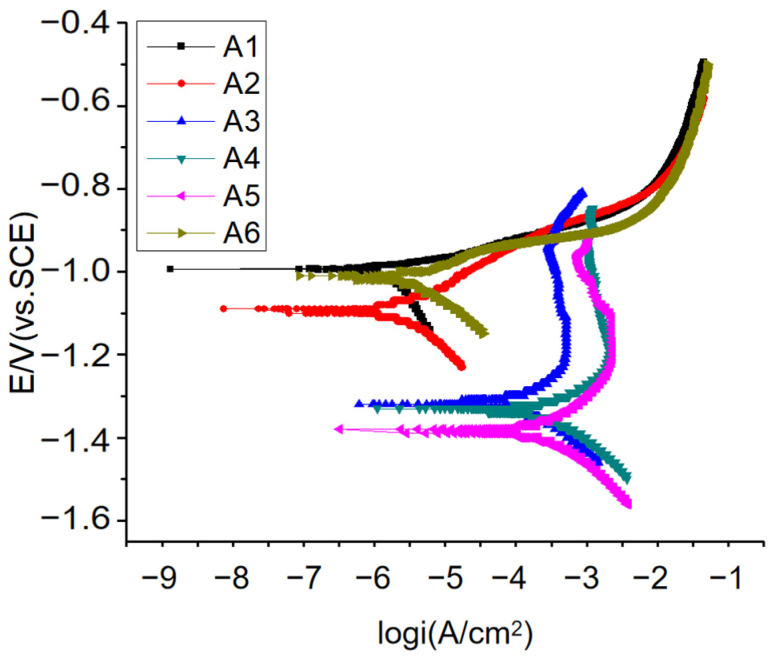
Polarization curves of the aluminum alloys.

**Figure 7 materials-17-00811-f007:**
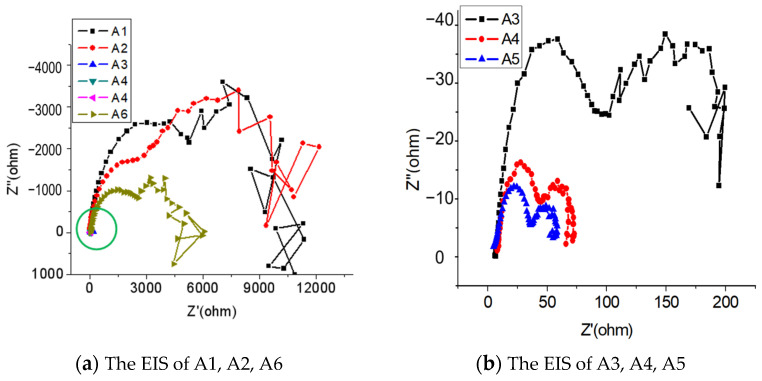
The EIS of the specimens.

**Figure 8 materials-17-00811-f008:**
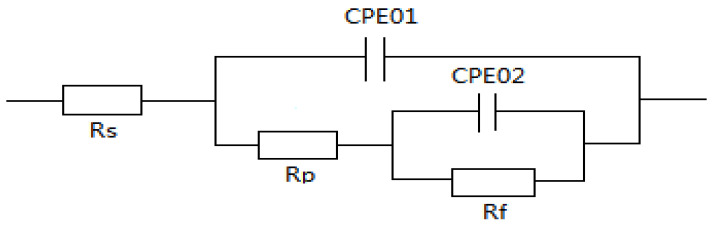
Equivalent circuit fitted to the EIS of the aluminum anode.

**Figure 9 materials-17-00811-f009:**
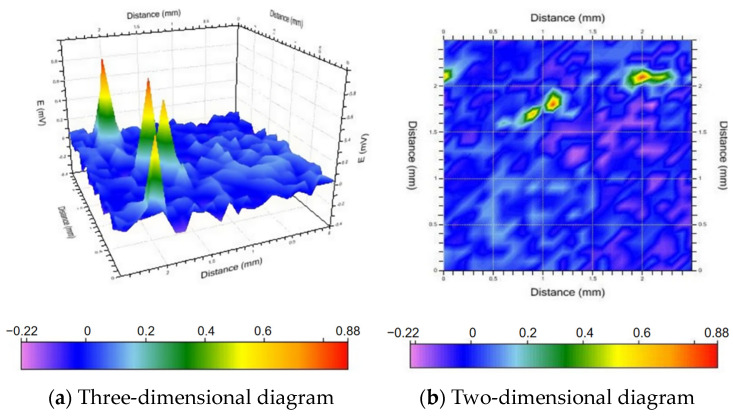
Voltage distribution diagram after 1.5 h of immersion.

**Figure 10 materials-17-00811-f010:**
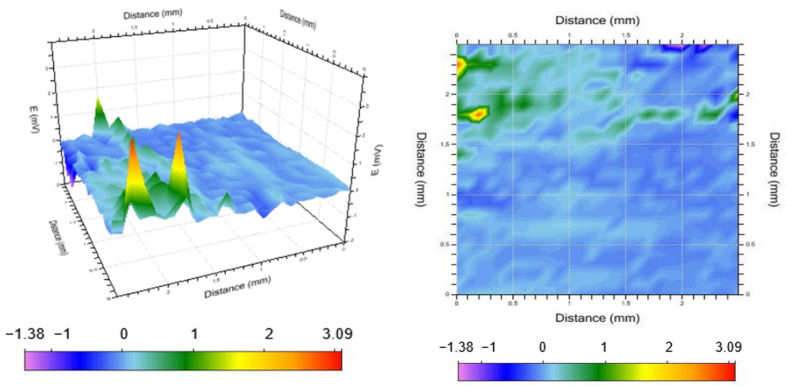
Voltage distribution after soaking for 4.5 h.

**Figure 11 materials-17-00811-f011:**
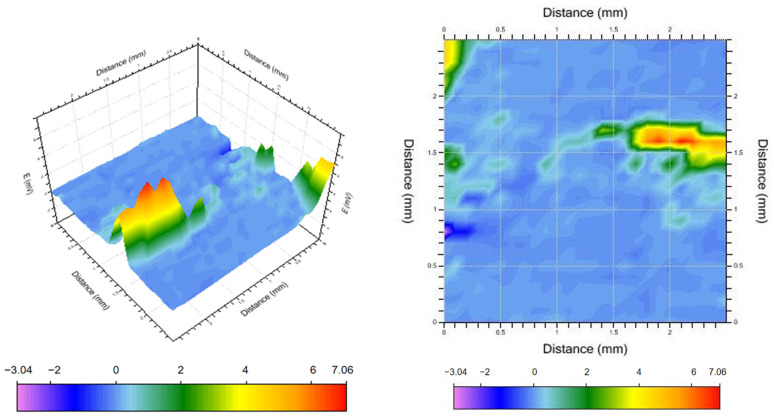
Voltage distribution after 5.5 h of soaking.

**Table 1 materials-17-00811-t001:** Anode composition formulations.

Serial Number	Pb (wt%)	Bi (wt%)	Ga (wt%)	In (wt%)	Al (wt%)
A1	5	0	0.01	0.025	Balance
A2	4	1	0.01	0.025	Balance
A3	3	2	0.01	0.025	Balance
A4	2	3	0.01	0.025	Balance
A5	1	4	0.01	0.025	Balance
A6	0	5	0.01	0.025	Balance

**Table 2 materials-17-00811-t002:** Working performance test results of sacrificial anode samples A1 to A6.

No.	Open-Circuit Potential (V)	Working Potential (V)	Actual Capacitance (A·h·kg^−1^)	Current Efficiency (%)
A1	−0.947	−0.987~−1.034	2241.59	78.82
A2	−1.063	−0.985~−1.027	2229.56	78.36
A3	−1.356	−0.75~−1.203	1050.72	36.91
A4	−1.366	−1.216~−1.313	835.54	29.34
A5	−1.438	−1.361~−1.493	1136.57	39.89
A6	−0.998	−0.977~−1.043	1173.19	41.16

**Table 3 materials-17-00811-t003:** Anode performance test results of sacrificial anode samples A1 to A6.

No.	Dissolved Morphology
A1	Pitted, pitting is more uniform, and the product is easy to fall off.
A2	Small pits, corrosion is more uniform, and the product is easy to fall off.
A3	Pockmarks, blisters, cracks, uneven erosion, and a small amount of product adherence.
A4	Blisters, cracks, grain detachment, uneven corrosion, and a small amount of product adherence.
A5	There are pits, islands, more uneven corrosion, and more product adhesion.
A6	Total corrosion, uniform corrosion, and a large amount of product adherence.

**Table 4 materials-17-00811-t004:** Tafel parameters of samples A1 to A6.

No.	Ba (mV)	Bc (mV)	Io (A/cm^2^)	Eo (Volts)	Corrosion Rate (mm/a)
A1	44.997	437.72	2.365 × 10^−6^	−0.99363	0.02583
A2	76.148	101.58	9.6624 × 10^−7^	−1.0931	0.010553
A3	542.12	231.54	0.00037519	−1.3181	4.0978
A4	232.11	231.11	0.00066046	−1.3321	7.2135
A5	204.31	180.79	0.00042815	−1.3849	4.6762
A6	44.59	100.48	2.0996 × 10^−6^	−1.0136	0.022932

**Table 5 materials-17-00811-t005:** Data table of the fitted EIS.

No.	R_S_ (ohm·cm^2^)	Y_01_ (S·sec^n^/cm^2^)	n_01_ (0 < n < 1)	R_P_ (ohm·cm^2^)	Y_02_ (S·sec^n^/cm^2^)	n_02_ (0 < n < 1)	R_f_ (ohm·cm^2^)
A1	4.06	1.1 × 10^−5^	0.825	6210	0.00015	1	4328
A2	2.72	1.8 × 10^−5^	0.984	2848	0.000144	0.31	7985
A3	6.29	6.5 × 10^−5^	0.836	81.77	0.0052	0.26	105
A4	8.26	0.01087	0.8	36.54	0.00529	0.8	23.28
A5	4.68	0.0278	0.689	34.73	0.0146	0.8	17.87
A6	9.59	2.2 × 10^−5^	0.801	2450	0.0411	0.79	2187

## Data Availability

Data are contained within the article.

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
