# Peer review of "Effect of Bi on the Performance of Al-Ga-In Sacrificial Anodes"

_materials, 2024, doi:10.3390/ma17040811_

Round 1

Reviewer 1 Report

Comments and Suggestions for Authors

Dear authors, please see the remarks presented in the attached review document.

Comments on the Quality of English Language

Dear authors, please see the remarks presented in the attached review document.

Reviewer 2 Report

Comments and Suggestions for Authors

This paper studied the effect of alloying elements (Pb, Bi, Ga, and In) on the electrochemical performance of Al sacrificial anodes. The paper includes a relatively large number of alloys (12); however, the results are not sufficiently discussed. Additionally, the paper has major deficiencies in style and language. As such, I cannot recommend it for publication. The following comments should be addressed:

1.The introduction is very short. There is no state-of-the-art review of status on the effect of alloying elements on the sacrificial anode performance. You must adequately explain what the benefits of individual elements are.

2.It is not clear if the percentage in Table 1 is at. % or wt. %.

3.I suppose that Al is the major element. The concentration of Al should be given as a balance instead of “margin”.

4.The entire materials and methods section reads like a recipe on how the experiments SHOULD BE done. It does not describe how the experiments WERE done. Furthermore, it includes many unnecessary details that should be omitted. An example is here (preparation section, lines 80-92): “Wash the floating dust…, remove the oil stain…, put the sample in a vacuum oven for an hour…, weigh it…, etc.” The section must be completely rewritten. You must rewrite it in the past tense and describe how the experiments were done. Include only the necessary details that will allow a specialist to repeat the experiments. Avoid writing recipes.

5.You must specify the reference electrode that was used during electrochemical experiments. Furthermore, it is not clear which potentiostat, scanning electron microscope, and SVET instrument were used during the analyses. You must specify the models.

6.The results are not organized in a logical sequence. For example, the corrosion morphology is presented first. The microstructure and chemical composition of the alloys must be presented prior to corrosion. Move the sections 3.3.1, .3.2 and 3.3.3 upfront. Afterward, electrochemical experiments should be presented (Sections 3.1.2, 3.2.1 and 3.2.2). Finally, the morphology of corrosion products should be studied (Sections 3.1.1 and 3.3.4).

7.Most of the SEM/EDS results are given for the last three samples only (A10-A12, Figs. 6-11). Why only these three? Why is the rest of the alloys omitted?

8.Corrosion results are summarized only in a simple table (Table 3). How did the OCP vary over time? Furthermore, you must include results for pure Al in the table to make the results comparable.

9.The Tafel curves are not analyzed (Fig. 3). You must obtain corrosion currents and corrosion potentials from the curves and compare them.

10.The EIS results are not analyzed (Fig. 4). You must present the equivalent circuits and obtain the electrical parameters from the measured curves.

11.You suggest that ZnAl2O4 might have been formed during corrosion (line 427). It would help to support the discussion of corrosion products by presenting the X-ray diffraction patterns.

12.It is not clear which alloys are presented in Figs. 12-14. Write it clearly in the figure caption.

13.The discussion of the results, in general, is poor and difficult to follow. The paper contains many unfinished sentences (lines 479, 488, 490, etc.). The authors often use expressive expressions (“aluminum anode is a “rising star” in sacrificial anode research, line 33). I recommend having your paper proof-read by native speaker before resubmission.

Comments on the Quality of English Language

The paper has major deficiencies in style and language:

The entire materials and methods section reads like a recipe on how the experiments SHOULD BE done. It does not describe how the experiments WERE done. Furthermore, it includes many unnecessary details that should be omitted. An example is here (preparation section, lines 80-92): “Wash the floating dust…, remove the oil stain…, put the sample in a vacuum oven for an hour…, weigh it…, etc.” The section must be completely rewritten. You must rewrite it in the past tense and describe how the experiments were done. Include only the necessary details that will allow a specialist to repeat the experiments. Avoid writing recipes.

The discussion of the results, in general, is poor and difficult to follow. The paper contains many unfinished sentences (lines 479, 488, 490, etc.). The authors often use expressive expressions (“aluminum anode is a “rising star” in sacrificial anode research, line 33). I recommend having your paper proof-read by native speaker before resubmission.

Reviewer 3 Report

Comments and Suggestions for Authors

1. Introduction must be elaborate with literature of similar previously published articles.

2. It is better to incorporate a table consist of comparitive results with similar work of other researchers. This will improve the introduction esction.

3. The authors specifically choosed  the design template, the total content of Pb+Bi is 5%, and the total content of Ga+In is 0.035%. Why? Is there any particular reason for that?

4. As per the line no 74, the sample is air cooled, then what about the oxidation? Because at higher tempearture materials are more prone fro oxidation. If they undergo oxidation during air cooling itself, then that will affect the corrosion studies.

Comments on the Quality of English Language

Many grammatical errors are present in the manuscript

Round 2

Reviewer 1 Report

Comments and Suggestions for Authors

Dear authors, I see major improvements of your manuscript. You answered clearly and satisfactory to all my review remarks. Your research paper can be published in the MDPI journal.

Reviewer 2 Report

Comments and Suggestions for Authors

The authors partially answered my previous comments. The paper has been slightly improved. However, major explanations and modifications are still necessary:

1.The Ga and In atomic fractions in the alloys were constant and relatively low (Table 1). Were these elements artificially introduced or were they just naturally present as impurities in aluminum?

2.Are the alloys A1-A6 commercial alloys? How were the alloys prepared?

3.Fig. 2 is incomplete. EDS spectra are invisible. Furthermore, a scale bar is missing.

4.In Table 2, it is written that the analyzed point in the A1 alloy contained 2.85 % rhenium (Re)?? How is it possible? Was this element really present in the alloy?

5.The A1 alloy is supposed to have 0 % Bi according to Table 1. How is it then possible that the analyzed EDS point spectra have 93 and 47 % Bi, respectively (Table 2)??

6.Figures 3, 4 and 5 do not have any scale bars.

7.There are huge differences (2 orders of magnitude) between the corrosion currents of the A3-A5 alloys on one hand and A1,A2 and A6 alloys on the other hand (Fig. 7, Table 5). A large difference is also found in the corrosion potentials. It is not clear why the A5 and A6 alloys were so different. Is the A6 alloy an outlier? Why? A discussion of the effects of Bi and Pb on the corrosion parameters is lacking in the manuscript.

8.In Fig. 8, there are EIS data for 3 alloys only (A1, A2, and A6) instead of all 6. Furthermore, the data in Fig. 8 indicate that the corrosion behavior of A6 is markedly different compared to those of A1 and A2. A major difference is also seen in Fig. 6. It sharply contradicts the polarization behavior where the A1, A2 and A6 alloys were comparable (Fig. 7, Table 5).

9.The authors discuss secondary phase formation at several places in the manuscript (p. 5, 6, etc.). It would help to inspect the alloys by XRD to confirm the phase constitution.

10.You should include the line numbers in the manuscript. It facilitates orientation and helps reviewers in indicating specific positions in the manuscript where modifications and corrections are necessary.
